# Progress in survival in renal cell carcinoma through 50 years evaluated in Finland and Sweden

**Kari Hemminki**[1,2]*, **Asta Försti**[3,4], **Akseli Hemminki**[5,6], **Börje Ljungberg**[7],
**Otto Hemminki**[5,8,9]

**1** Biomedical Center, Faculty of Medicine and Biomedical Center in Pilsen, Charles University in Prague, Pilsen, Czech Republic, **2** Division of Cancer Epidemiology, German Cancer Research Center (DKFZ), Heidelberg, Germany, **3** Hopp Children's Cancer Center (KiTZ), Heidelberg, Germany, **4** Division of Pediatric Neurooncology, German Cancer Research Center (DKFZ), German Cancer Consortium (DKTK), Heidelberg, Germany, **5** Cancer Gene Therapy Group, Translational Immunology Research Program, University of Helsinki, Helsinki, Finland, **6** Comprehensive Cancer Center, Helsinki University Hospital, Helsinki, Finland, **7** Department of Surgical and Perioperative Sciences, Urology and Andrology, Umeå University, Umeå, Sweden, **8** Department of Urology, Helsinki University Hospital and University of Helsinki, Helsinki, Finland, **9** Division of Urologic Oncology, Department of Surgical Oncology, Princess Margaret Cancer Center, University Health Network and University of Toronto, Toronto, Ontario, Canada

* K.Hemminki@dkfz.de

**Data Availability Statement:** The source data is publicly accessible at https://NORDCAN.iarc.fr/en/database#bloc2. We collected information from this database and organized it to tables and figures

## Abstract

Global survival studies have shown favorable development in renal cell carcinoma (RCC) treatment but few studies have considered extended periods or covered populations for which medical care is essentially free of charge. We analyzed RCC survival in Finland and Sweden over a 50-year period (1967–2016) using data from the NORDCAN database provided by the local cancer registries. While the health care systems are largely similar in the two countries, the economic resources have been stronger in Sweden. In addition to the standard 1- and 5-year relative survival rates, we calculated the difference between these as a measure of how well survival was maintained between years 1 and 5. Relative 1- year survival rates increased almost linearly in both countries and reached 90% in Sweden and 80% in Finland. Although 5-year survival also developed favorably the difference between 1- and 5-year survival rates did not improve in Sweden suggesting that the gains in 5-year survival were entirely due to gains in 1-year survival. In Finland there was a gain in survival between years 1 and 5, but the gain in 1-years survival was the main contributor to the favorable 5-year survival. Age group specific analysis showed large survival differences, particularly among women. Towards the end of the follow-up period the differences narrowed but the disadvantage of the old patients remained in 5-year survival. The limitations of the study were lack of information on performed treatment and clinical stage in the NORDCAN database. In conclusion, the available data suggest that earlier diagnosis and surgical treatment of RCC have been the main driver of the favorable change in survival during the past 50 years. The main challenges are to reduce the age-specific survival gaps, particularly among women, and push survival gains past year 1.

shown in the publication exactly in the form collected from NORDCAN.

**Funding:** European Union's Horizon 2020 research and innovation programme, grant No 856620 (Chaperon), to KH, Jane and Aatos Erkko Foundation (6-5900-29), Sigrid Juselius Foundation (63-4925-56), Finnish Cancer Organizations (6-5156-32), Biomedicum Helsinki Foundation (73604201), University of Helsinki (797011008), Helsinki University Central Hospital (TYH2019215), Novo Nordisk Foundation (0058602), Päivikki and Sakari Sohlberg Foundation (10-6623-11), all to AH. Funding agencies had no role in the study.

**Competing interests:** The authors have read the journal's policy and have the following competing interests: AH is shareholder in Targovax ASA. AH is also employee and shareholder in TILT Biotherapeutics Ltd. Other authors declared no conflict of interest. This does not alter our adherence to PLOS ONE policies on sharing data and materials. There are no patents, products in development or marketed products associated with this research to declare.

## Introduction

Survival in many cancers, including renal cell carcinoma (RCC), has improved over the past years in the developed countries [1]. Although the underlying data appear undisputed, the reasons for the favorable development in survival have many interpretations. The key role of clinical randomized trials in selecting the optimal treatment is universally emphasized in enabling these success stories, but for many cancer patients the 'real world' cure circumstances may be far from the selected patient populations and controlled treatment protocols of the clinical trials. Many survival studies cover relatively short periods which do not allow assessment of the survival experience over decenniums, which would be important to understand the factors influencing the 'real world' survival trends [1–3].

RCC is characterized by male excess, ranging from 2- to 4-fold, and known risk factors of smoking, overweight and obesity, and germline mutations in specific genes [4]. In the developed world there has been a general increase in incidence but it has stabilized or slightly declined in some countries [4, 5]. Early detection and improvements in treatment have contributed to positive trends in RCC survival [6]. Novel imaging technologies, detailed under Methods 'Diagnostics and treatment for RCC', have resulted in frequent incidental detection of tumors, which tend to be smaller and detected earlier than symptomatic tumors. Standard treatment for RCC has been surgery with a trend during the recent years towards minimally invasive techniques. After 2006, antiangiogenic drugs have largely replaced cytokine treatments in metastatic RCC (mRCC), however their impact in survival has been debatable [6, 7].

We assessed RCC survival experience in Finland and Sweden over a period of 50 years. For a 'real world' experience these two countries are examples of practically free-of-charge medical care to the population at large receiving similar treatments for RCC [8, 9]. Survival in most cancers in Finland and Sweden has been above the European mean but for 5-year survival in kidney cancer the two countries were below the European mean in years 2000 to 2007 [2]. With analysis of the survival patterns in the NORDCAN database we estimated factors underlying improvements in survival in RCC over a 50-year period.

## Methods

The data used originate from the NORDCAN database which is a compilation of data from the high-level Nordic cancer registries as described [10] (https://NORDCAN.iarc.fr/en/database#bloc2). These registries are presented in detail by Pukkala and coworkers [11]. Data on the Finnish and Swedish RCC patients, diagnosed between 1967 and 2016, were extracted from NORDCAN. RCC patients included also Wilms tumor patients who accounted for less than 1.5% of all; however, for the age group below 50 years they accounted for 12% of male and 20% of female patients. Because 98.5% of cancers are RCC we consider justified to use this terminology in the title. All survival data are 'relative survival' which is defined as the ratio of the observed survival in the group of patients compared to the survival expected in the general population, adjusted for sex, age and calendar time at the time of diagnosis. Survival data were available from 1967 onwards and the analysis was based on the cohort survival method for the first nine 5-year periods from 1964–2011, and a hybrid analysis combining period and cohort survival in the last period 2012–2016, as detailed [12, 13]. The Finnish and Swedish life tables were used to calculate the expected survival. For statistical assessment of survival data 95% confidence intervals (CIs) were provided for each 5-year survival percentage. Statistical significance was called when 95%CIs for two survival figure did not overlap.

We calculated also a difference in survival percent between year 1 and year 5 as a measure on how well survival is maintained between years 1 and 5. A small difference indicates high survival between years 1 and 5 after diagnosis.

## Diagnostics and treatment for RCC

The development of the tumor-node-metastasis (TNM) classification and staging system has been important for the standardization of diagnostics and treatment in cancers since 1958 [14]. Over the years, the diagnostic arsenal has increased to include ultrasound (US) and computed tomography (CT). A Finnish study defined the diagnostic periods: pre-CT and pre-US era (1964–1979), US era (1980–1988) and CT era (1989–1997) [15]. The proportion of tumors 3 cm or smaller of all tumors in these periods were 4.4, 9.8 and 16.6%. A more recent publication from Sweden reported 29% T1a (<4cm) during 2005–2013 [16]. The median tumor size at detection decreased from 60 mm in 2005 to 55 mm in 2013.

Surgery has been the traditional treatment for RCC and has remained so until today. For localized RCC, surgery has a curative intent. Especially with small renal masses, the 5-year cancer specific survival is excellent (>97%) [17]. The general trend during the last decades has been towards treatments with partial nephrectomy and also minimally invasive treatments. These surgical trends have suggested equal oncological outcomes with less adverse events [18]. Further, the increased detection of small renal masses has contributed to this trend. In the "cytoreductive" mRCC surgery the aim has been for palliation and/or for prolongation of survival.

mRCC has poor prognosis. Among 223 RCC patients treated in a Swedish university hospital between 1982 and 1993, 44.4% had distant metastases (including 31 palliative patients) and 19.7% local metastases [19]. All but palliative patients were nephrectomized and their tumor sizes were 87 mm for aneuploidy samples (72%) and 62 mm for the remaining diploid samples. The mean survival was 23 months for deaths in RCC and 33 months for deaths in 'intercurrent diseases' [19]. In the same hospital, 74% of 106 mRCC patients treated between 1982 and 1999 were nephrectomized [20]. All but 15 of these patients received at least one other treatment such as: medroxyprogesterone acetate or tamoxifen (23 patients); interferon and/or interleukin-2 (21 patients); excision of metastases (17 patients); radiation therapy (34 patients). Median survival was 7 months [20].

According to the National Swedish Kidney Cancer Register, covering years 2005 and 2010, more than half of patients presented with T1 tumor and the mean tumor size decreased from 70 to 64 mm; the frequency of mRCC decreased from 22% to 15% [21]. The use of preoperative chest CT increased from 59% to 84%. In total, 76% of patients were treated with curative intent, and of these 82% underwent radical nephrectomy, 14% partial nephrectomy and 4% cryotherapy or radiofrequency ablation. In patients with mRCC, 55% underwent cytoreductive nephrectomy. Among patients diagnosed without metastases, 20% developed metastasis or local recurrence within 5 years [22]. Half of patients were treated with systemic oncological treatment, 17% underwent metastasectomy, 3% resection of local recurrences and 27% no tumor-specific tumor treatment. Targeted therapies for mRCC were approved by European Medical Agency in 2006 and were widely taken to use in mRCC patients younger than 75 years (7).

## Results

The NORDCAN database included 0.37 million male and 0.48 million female cancers for Finland, and 1.01 million male and 0.94 million female cancers for Sweden, excluding non-melanoma skin cancer, for years 1967 to 2016. In Finland, male RCC numbered 16,576 (median age at diagnosis 67 years) compared to 13,280 female RCC (70 years); the related numbers for Sweden were 30,597 (67 years) and 21,110 (69 years). During the follow-up time there was a shift in diagnostic age from period 1967–71 to 2012–16 for Finnish patients only, for men from 62 to 68 years and women from 66 to 72 years.

**Table 1. Relative 1-year and 5-year survival (%) and their Difference (Diff) in Finland and Sweden.**

| | Finland | | | | | Sweden | | | | |
|---|---|---|---|---|---|---|---|---|---|---|
| MEN | 1y | 95%CI | 5y | 95%CI | Diff | 1y | 95%CI | 5y | 95%CI | Diff |
| 1967–1971 | 51 | [46;57] | 27 | [22;34] | **24** | 44 | [42;45]* | 27 | [25;29] | **17** |
| 1972–1976 | 53 | [49;58] | 30 | [26;36] | **23** | 49 | [47;51]* | 31 | [29;33]* | **18** |
| 1977–1981 | 58 | [54;62] | 35 | [30;40] | **23** | 56 | [54;59] | 37 | [35;40] | **19** |
| 1982–1986 | 59 | [56;62]* | 36 | [33;40]* | **23** | 60 | [58;62]* | 40 | [38;42] | **20** |
| 1987–1991 | 67 | [64;69]* | 48 | [45;52]* | **19** | 65 | [63;67]* | 44 | [42;47] | **21** |
| 1992–1996 | 73 | [71;75] | 54 | [51;58] | **19** | 70 | [68;72]* | 49 | [47;51] | **21** |
| 1997–2001 | 74 | [72;76] | 58 | [55;61] | **16** | 71 | [69;73]* | 52 | [50;55]* | **19** |
| 2002–2006 | 76 | [73;78] | 60 | [57;63] | **16** | 77 | [76;79]* | 60 | [58;63]* | **17** |
| 2007–2011 | 77 | [75;79] | 61 | [59;64] | **16** | 85 | [84;87]* | 69 | [67;72] | **16** |
| 2012–2016 | 79 | [78;81] | 64 | [62;66] | **15** | 90 | [89;91] | 72 | [70;74] | **18** |
| WOMEN | | | | | | | | | | |
| 1967–1971 | 52 | [48;57] | 30 | [27;35] | **22** | 50 | [48;52] | 34 | [32;37] | **16** |
| 1972–1976 | 57 | [53;62] | 36 | [32;40] | **19** | 53 | [51;55]* | 37 | [34;39] | **16** |
| 1977–1981 | 61 | [57;65] | 38 | [34;43] | **23** | 59 | [57;62] | 40 | [37;42] | **19** |
| 1982–1986 | 66 | [63;69] | 46 | [43;50] | **20** | 61 | [59;64] | 43 | [41;46] | **18** |
| 1987–1991 | 71 | [69;74] | 54 | [50;57] | **17** | 65 | [63;67]* | 45 | [43;48]* | **20** |
| 1992–1996 | 73 | [70;75]* | 59 | [56;62] | **14** | 72 | [70;74] | 54 | [52;57] | **18** |
| 1997–2001 | 78 | [76;80] | 62 | [59;65] | **16** | 75 | [73;77] | 57 | [55;60] | **18** |
| 2002–2006 | 79 | [77;81] | 65 | [62;68] | **14** | 78 | [76;80]* | 62 | [59;64]* | **16** |
| 2007–2011 | 82 | [80;84] | 67 | [65;69] | **15** | 87 | [85;89]* | 72 | [69;74] | **15** |
| 2012–2016 | 82 | [81;84] | 68 | [66;70] | **14** | 89 | [88;91] | 74 | [72;76] | **15** |

* Indicate that the 95%CIs do not overlap between the marked 5-year period and the next one.

Relative 1-year and 5-year survival rates for RCC for Finland and Sweden is shown in **Table 1**. The male 1-year survival increased constantly, from 51% (1967–1971) to 79% (2012–2016) for Finnish men and from 44% to 90% for Swedish men. For women the increases were from 52 to 82% (Finland) and from 50 to 89% (Sweden). The male 5-year survival rate increased from 27to 64% (Finland) and from 27 to 72% (Sweden). For women the increases were from 30 to 68% and from 34 to 74%, respectively.

In **Table 1**, the stars after the survival percent indicate a significant increase in survival (i.e., 95%CIs were non-overlapping) between this and the next period. Among Finnish men large gains took place in the period from 1982 to 1991 while for Swedish men and women many periods were favorable for 1-year survival. Overall, 1-year survival increased in every 5-year period in both countries and sexes (except among Finnish women during 2012–2016). It is notable in the country comparison that after 1997 no single increase between 5-year periods were significant in Finland compared to many significant increases for Swedish men and women. The consequence was that 1- and 5-year survival for Swedish men and women was significantly better compared to Finland since 2007.

**Table 1** shows also the difference between 1- and 5-year survival rate in percent units (% units). In Finland the difference decreased for men from 24 to 15% units, and for women from 22 to 14% units. In Sweden, the difference was initially lower for both sexes than in Finland but it then increased and again declined to the initial level. The survival data are plotted in **Fig 1**A (Finland) and 1B (Sweden) illustrating the declining difference between the survival bars in Finland, opposite to Sweden.

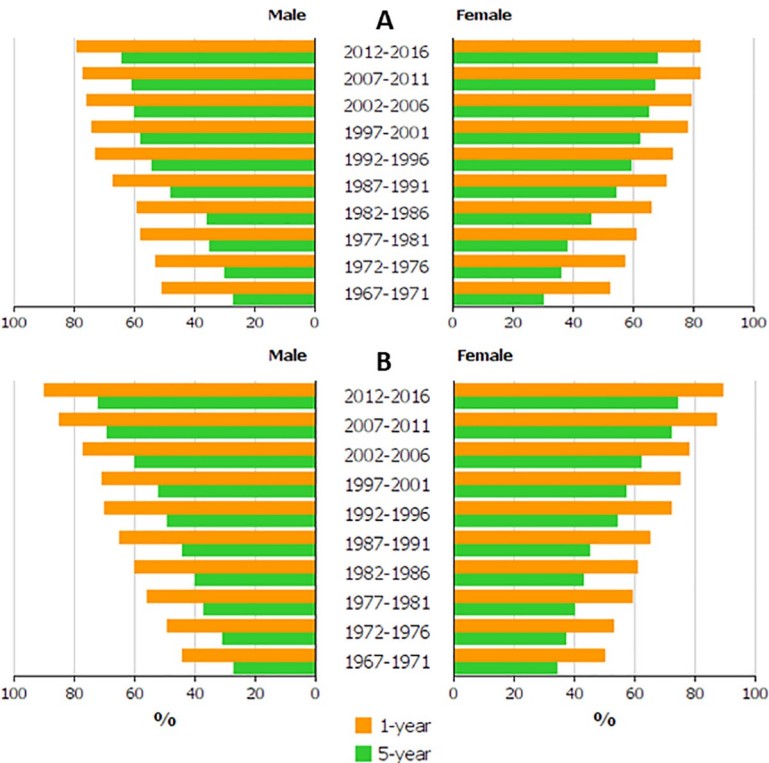

**Fig 1.** Relative 1-year (yellow bars) and 5-year (green bars) survival in RCC among Finnish (A) and Swedish (B) men and women.

Trend in relative survival is shown for Finnish and Swedish men and women in **Fig 2**. The curves for 1-year survival (**Fig 2A**) crossed, starting lower for Swedish men but ending highest. The Swedish graphs were linear, for Finnish patients they showed a minor degree of concavity. The graphs for 5-years survival (**Fig 2B**) started highest for Swedish and Finnish female patients and ended with Swedish women and men on top.

Age-specific 1- and 5-year relative survival for RCC in Finnish and Swedish men shows different trends for the two countries (**Fig 3**). While in Sweden the age group specific survival rates were hugely different in the early period, they constantly narrowed in the course of time. In Finland the opposite was the case and age group specific survival differences remained wide with time; 1-year survival for men aged 80–89 years lagged 30% units below the youngest patient group.

Age-specific survival for women showed even a larger age group specific difference. Some narrowing of the curves was seen among the Swedish women, while no such phenomenon was noted in Finland. (**Fig 4**).

## Discussion

The almost linear 1-year survival graphs for Finnish and Swedish patients over the 50 years period suggest that no single event in the care of RCC influenced the improvement of the survival. Instead the constant increase in survival support the role of gradual improvements and penetration into the national health care. A little more heterogeneity was observed in 5-year survival but it is instructive to consider how much of 5-year survival was driven by increase in 1-years survival, measured by the difference between these two (**Table 1**). The results were

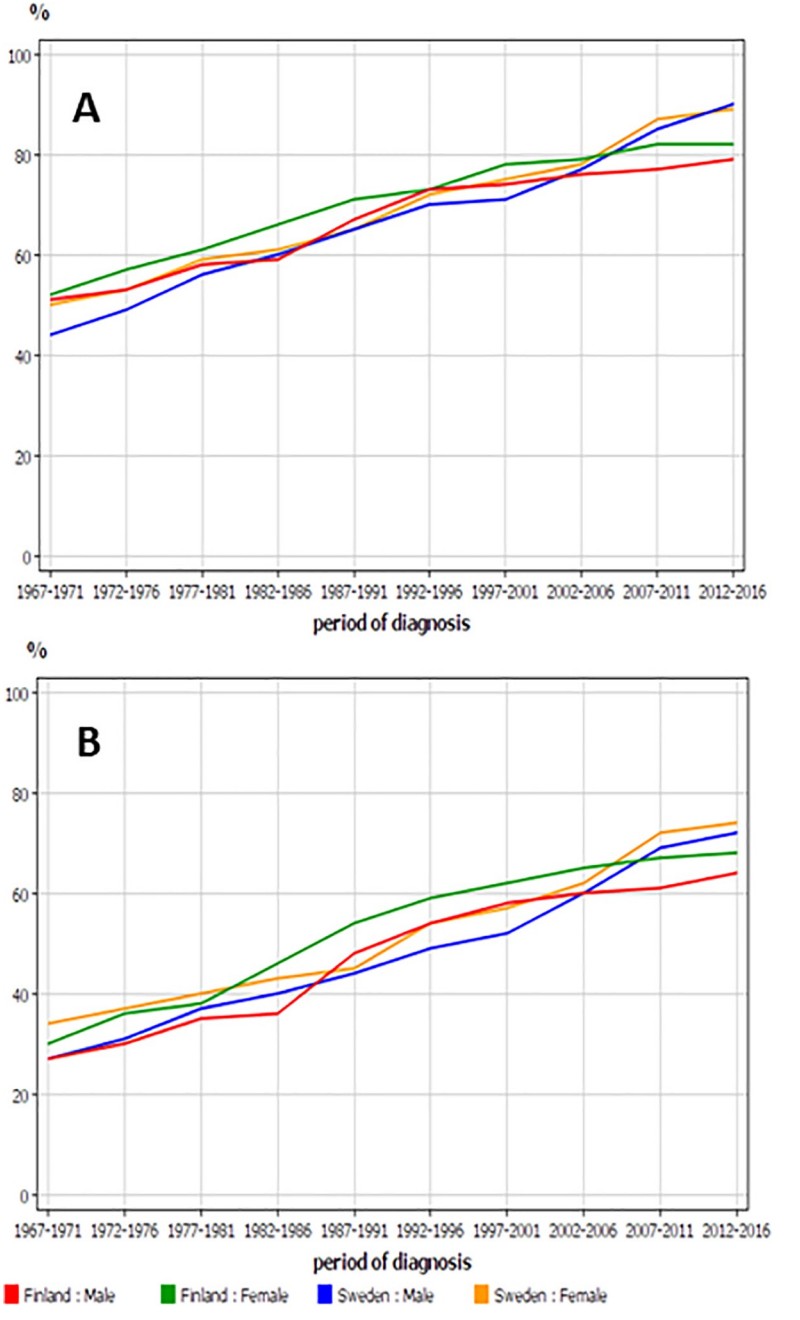

**Fig 2.** Relative 1-year (A) and 5-year (B) survival in RCC among Finnish and Swedish men and women.

quite different for Finland and Sweden but were consistent between the sexes in each country. In Finland, the difference decreased over time which indicates that the gap between 1- and 5-year survival narrowed, but nevertheless the larger part of the gains in 5-years survival was due to gains in 1-year survival. In Sweden, no survival benefit was achieved between years 1 and 5.

Patients surviving one year include those who were cured with treatment (e.g., surgery) and those who are alive with disease but will eventually succumb to metastatic or recurrent disease. Patients surviving 5 years are cured because few patients would be expected to survive that

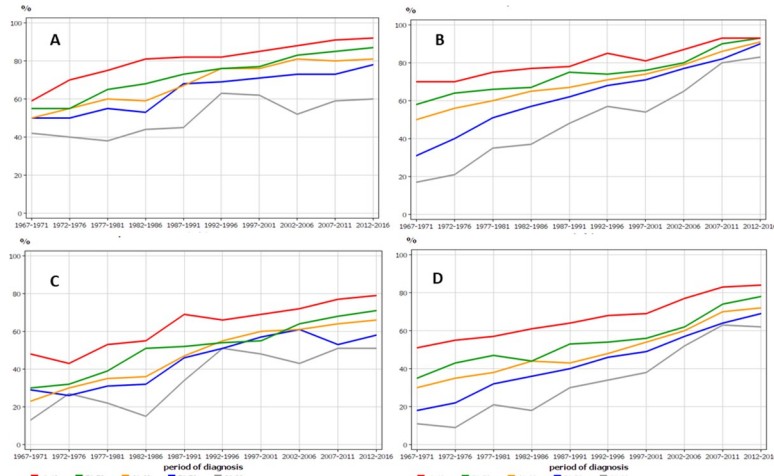

**Fig 3.** Age-specific 1-year relative survival for RCC in Finnish (A) and Swedish (B) men, and the related 5-year survival in the same male populations (C, D).

long with disease, although RCC may recur even after more than 5 years after treatment [22]. RCC is an "immunogenic" tumor where regression, including metastases, has been proposed to occur in up to 1% of patients [23]. Also, effective immunotherapies include high dose interleukin 2 and checkpoint inhibitors, but the latter are too new to impact the data reported here (nivolumab was approved in 2015) [7].

Interpretation of 1- and 5-years survival data is that the narrower the difference, the better patients are diagnosed early and treated with curative options, which in most cases indicates surgery. In Sweden, the difference increased towards period 1987–1991 and then declined to the starting level; after 2002 no change took place. A Swedish study focusing on the possible survival effects of antiangiogenic drugs reported that in period 2009–2012 the median survival in mRCC was 18 months, but the present analysis shows no independent increase in 5-year

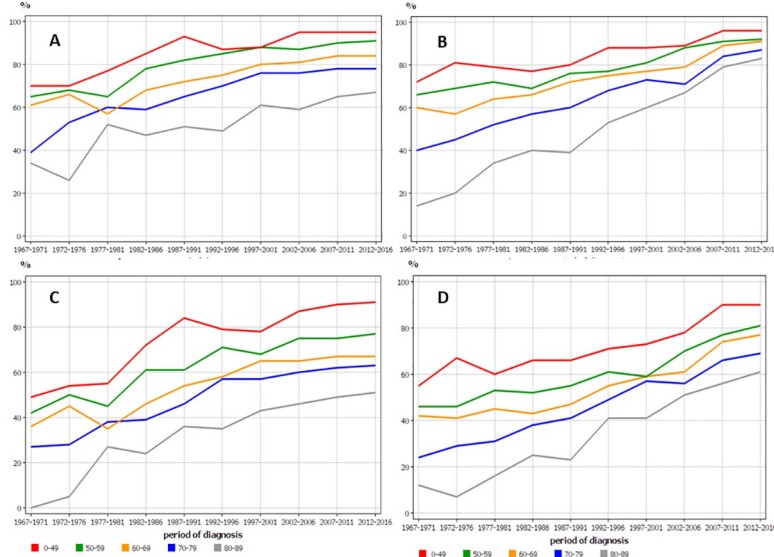

**Fig 4.** Age-specific 1-year relative survival for RCC in Finnish (A) and Swedish (B) women, and the related 5-year survival in the same female populations (C, D).

survival [24]. Thus, in Sweden the increase in 5-year survival was entirely driven by increase in 1-year survival.

Translating these findings into past clinical reality is difficult because only a few, single-center studies are available with stage data. The Swedish data from years 1982 and 1993 showed that 44.4% of the diagnosed patients had distant metastases and the mean tumor size was around 80 mm [19]. National data from years 2005 to 2010, showed that the proportion of mRCC had dropped to 15% and the mean tumor size to 64 mm towards the end of this period [21]. These data suggest that there was a dramatic stage migration toward early-stage cancers in two decades. An interesting corollary is the number of CT instruments in Sweden [25]. First CTs were acquired in the early 1970s, and the cumulative number of installed units reached to 15 by 1979, 85 by 1989 and 125 by 1999.

The reported survival data for mRCC was 23 months in the Swedish data from years 1982 to 1993 which would fit the improvements observed in survival [19]. In the 1970s, almost half of RCC patients (those with distant and local advanced disease) died during the first year (1-year survival ~50%). Improving imaging technologies were able to find increasingly early stages and towards 2010 mRCC accounted for only 15% of all RCC (1-year survival ~85% in Sweden, slightly lower in Finland). Maybe more active surgery has contributed, as well as alertness about RCC in the primary care and population in general. The Finnish experience in finding smaller tumors with improving imaging was similar to the Swedish one [15].

The historical analysis should also consider age group specific survival, which showed surprisingly large differences, particularly among women. In the first observation period (1967–73), 5-year survival among Finnish women aged 80–89 was zero (**Fig 4C**)! For their Swedish counterparts it was barely over 10%. Their contemporary colleagues aged below 50 survived at 50%. The 1-year survival for men and women (particularly in Sweden) in the last period (2012–16) shows that surgery is able to effectively treat both young and old with very small survival differences (**Figs 3B, 4B**). The large survival disadvantage of the old patients, particularly in the early periods, were likely to be due to late detection and conservative treatments. Why the disadvantage persists among women even in the last period remain unclear.

A likely explanation for the slow development in survival in Finland starting in the 1990s is economic (**Fig 2**). Finland faced a significant economic crisis and its GDP per capita decreased from 28,400 to 17,600 USD in a few years and it took over ten years to surpass (https://www. macrotrends.net/countries/FIN/finland/gdp-per-capita). Cost savings involved also the heath care, including CT and US imaging, and most new machinery purchases were postponed. These measures made prioritizing an important factor and it seems that the elderly (females) were probably neglected.

The strengths of the study are that we have data from two countries with practically free medical care offered to the 'real world' population at large, and covered by nation-wide cancer registries of high-quality. The weaknesses are that the data are ecological and no individual level treatment or care data were available. This applies also to the overview of 'Diagnostics and treatment for RCC' described in Methods. Lack of stage data in the NORDCAN database does not allow inclusion of this variable which is an important predictor of survival [18].

As surgery is the overwhelming treatment modality, our discriminatory power for auxiliary treatment modalities, such as systemic oncological therapy is low. This applies also to the period of antiangiogenic drugs starting from 2006 onwards. However, the difference between 1-and 5-year survival did not show evidence for independent improvement in 5-year survival after year 2006 [7].

In conclusion, the present results show that 1-year survival in RCC has increased almost linearly in Finland and Sweden during the past 50 years, suggesting a gradual improvement without any major single break-throughs. The decrease in the proportion of mRCC and tumor

size at diagnosis as well as the growing incidence of incidental diagnosis suggests improvements in earlier diagnosis, which has probably been the main driver of the favorable survival. It seems highly plausible that the increasing amounts of CT and US imaging has led to earlier detection, and surgical treatments of RCC explaining most of the improved clinical results. The favorable survival is entirely limited to year 1 in Sweden, and largely also in Finland. The large age-group specific survival differences were probably not dependent on the therapeutic effectiveness of surgery but conservative approach in its use, in addition to delayed diagnosis. The main challenges are to reduce the age-specific survival gap, particularly among women, and push survival gains past year 1. More active diagnostic and surgical approaches seem to make a difference–special interest should be dedicated to the elderly and to females.

## Author Contributions

**Conceptualization:** Kari Hemminki, Otto Hemminki.

**Formal analysis:** Kari Hemminki, Otto Hemminki.

**Investigation:** Asta Försti, Akseli Hemminki, Börje Ljungberg.

**Project administration:** Kari Hemminki.

**Supervision:** Otto Hemminki.

**Validation:** Asta Försti, Akseli Hemminki, Börje Ljungberg.

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
