## [Decision Letter · Decision Letter 0]

20 Apr 2021

PONE-D-21-06223

PROGRESS IN SURVIVAL IN RENAL CELL CARCINOMA THROUGH 50 YEARS EVALUATED IN FINLAND AND SWEDEN

PLOS ONE

Dear Dr. Hemminki,

Thank you for submitting your manuscript to PLOS ONE. After careful consideration, we feel that it has merit but does not fully meet PLOS ONE’s publication criteria as it currently stands. Therefore, we invite you to submit a revised version of the manuscript that addresses the points raised during the review process.

We look forward to receiving your revised manuscript.

Kind regards,

Wen-Wei Sung, M.D., Ph.D.

Academic Editor

PLOS ONE

Journal Requirements:

To comply with PLOS ONE submission guidelines, in your Methods section, please provide additional information regarding your statistical analyses. For more information on PLOS ONE's expectations for statistical reporting, please see https://journals.plos.org/plosone/s/submission-guidelines.#loc-statistical-reporting.

We note that you have stated that you will provide repository information for your data at acceptance. Should your manuscript be accepted for publication, we will hold it until you provide the relevant accession numbers or DOIs necessary to access your data. If you wish to make changes to your Data Availability statement, please describe these changes in your cover letter and we will update your Data Availability statement to reflect the information you provide.

Thank you for stating the following in the Competing Interests section):

A.H. is shareholder in Targovax ASA. A.H. is employee and shareholder in TILT Biotherapeutics Ltd. Other authors declared no conflict of interest.

We note that one or more of the authors have an affiliation to the commercial funders of this research study : Targovax ASA and  TILT Biotherapeutics Ltd

4a. Please provide an amended Funding Statement declaring this commercial affiliation, as well as a statement regarding the Role of Funders in your study. If the funding organization did not play a role in the study design, data collection and analysis, decision to publish, or preparation of the manuscript and only provided financial support in the form of authors' salaries and/or research materials, please review your statements relating to the author contributions, and ensure you have specifically and accurately indicated the role(s) that these authors had in your study. You can update author roles in the Author Contributions section of the online submission form.

4b. Please also provide an updated Competing Interests Statement declaring this commercial affiliation along with any other relevant declarations relating to employment, consultancy, patents, products in development, or marketed products, etc. 

Reviewers' comments:

Reviewer's Responses to Questions

**Comments to the Author**

1. Is the manuscript technically sound, and do the data support the conclusions?

Reviewer #1: Partly

Reviewer #2: Yes

Reviewer #3: Partly

Reviewer #4: Yes

Reviewer #5: Yes

2. Has the statistical analysis been performed appropriately and rigorously? 

Reviewer #1: N/A

Reviewer #2: Yes

Reviewer #3: Yes

Reviewer #4: I Don't Know

Reviewer #5: Yes

3. Have the authors made all data underlying the findings in their manuscript fully available?

Reviewer #1: Yes

Reviewer #2: Yes

Reviewer #3: Yes

Reviewer #4: Yes

Reviewer #5: Yes

4. Is the manuscript presented in an intelligible fashion and written in standard English?

Reviewer #1: No

Reviewer #2: Yes

Reviewer #3: Yes

Reviewer #4: No

Reviewer #5: Yes

5. Review Comments to the Author

Reviewer #1: The authors used a database information to show the survival trend of kidney cancer in Finland and Sweden. It is an interesting observation study to see the improvement of survival in history. I have several comments as below.

1. Page 6, first paragraph, for treatment of mRCC, tyrosin kinase inhibitors are the major weapon since 2005.

2. Please dress the limitations of this study since all the inference were according to past literature, not this data.

Reviewer #2: This study provide important information regarding survival of RCC.

minor comment:

1: please discuss in the era of target therapy ( like sunitinib , other TKI) and immunotherapy , how does this novel agents influence the survival of metastatic RCC in Finland and Sweden?

Did the NORDCAN database included these information?

Reviewer #3: Hemminki et al., evaluated 50 years of survival data on RCC in Finland and Sweden.

In this descript analyses of data from two different countries both with free medical care show that in Finland there is an improvement in difference between 1- and 5- year survival data and such observation is missing in Sweden.

Comment: Authors documented reasonable explanations for the observed differences between Finland and Sweden and accounted for less healthcare spending in Finland during economic crisis. This economic crisis may be in the starting of 1990s. So comments need to be made whether the differences in survival after 1990s is significant between these two countries.

Reviewer #4: This manuscript presents valuable data which confirms the current knowledge regarding the natural history of renal cell carcinoma.

My comments are as follows:

The manuscript needs minor revision in terms of scientific writing.

The title talks about renal cell carcinoma but as far as I understood you have included Wilms tutor patients. I think one should be changed, either title or the analysis.

I honestly did not understand the purpose of section " Diagnostics and treatment for RCC". The fact that you mentioned this after methods part is a bit confusing.

Reviewer #5: Studies on Renal Cell Carcinoma survival till date has shown favorable survival outcomes. However, these studies have been performed over limited periods of time. In this manuscript, authors examined survival rates in RCC over an extended period of time in Swedish and Finland populations. They analyzed RCC survival in Finland and Sweden (with similar health care systems) over a 50-year period (1967-2016) using data from the NORDCAN database provided by the local cancer registries.

In addition to the standard 1- and 5-year relative survival rates, they calculated the difference between 1 year and 5-year survival as a measure of how well survival was maintained between years 1 and 5. While relative 1-

year survival rates increased linearly in both countries and reached 90% in Sweden and 80% in

Finland, 5-year survival also developed favorably. The difference between 1- and 5-year survival rates did not improve in Sweden suggesting that the gains in 5-year survival were owing to gains in 1-year survival. In Finland there was a gain in survival between years 1 and 5, but the gain in 1-years survival was the main contributor to the favorable 5-year survival. The authors conclude that earlier diagnosis and surgical treatment of RCC is the main driver of the favorable change in survival during the past 50 years. This study reports novel findings. Though due to limited data available in NORDCAN database, there was lack of information on performed treatments and clinical stage, precluding the analyses of important potential variables.

6. PLOS authors have the option to publish the peer review history of their article (what does this mean?). If published, this will include your full peer review and any attached files.

Reviewer #1: No

Reviewer #2: **Yes: **Shu-Pin Huang

Reviewer #3: No

Reviewer #4: No

Reviewer #5: No

---

## [Author Response · Author response to Decision Letter 0]

4 May 2021

REBUTTAL PONE-D-21-06223

Comments to the Author

We thank the reviewers for clarifying comments. Our changes are highlighted in the manuscript. 

As to a PONE general comment no. 2 we added statistical analysis details in Methods, bottom p. 3. 

Reviewer #1: The authors used a database information to show the survival trend of kidney cancer in Finland and Sweden. It is an interesting observation study to see the improvement of survival in history. I have several comments as below.

1. Page 6, first paragraph, for treatment of mRCC, tyrosin kinase inhibitors are the major weapon since 2005.

>>> Addition on p. 4. 

2. Please dress the limitations of this study since all the inference were according to past literature, not this data.

>>> Added to other limitations, p. 7

Reviewer #2: This study provide important information regarding survival of RCC.

minor comment:

1: please discuss in the era of target therapy ( like sunitinib , other TKI) and immunotherapy , how does this novel agents influence the survival of metastatic RCC in Finland and Sweden?

Did the NORDCAN database included these information?

>>> Addition on p. 3 and 6. 

Reviewer #3: Hemminki et al., evaluated 50 years of survival data on RCC in Finland and Sweden.

In this descript analyses of data from two different countries both with free medical care show that in Finland there is an improvement in difference between 1- and 5- year survival data and such observation is missing in Sweden.

Comment: Authors documented reasonable explanations for the observed differences between Finland and Sweden and accounted for less healthcare spending in Finland during economic crisis. This economic crisis may be in the starting of 1990s. So comments need to be made whether the differences in survival after 1990s is significant between these two countries.

>>> Significance was described in the new paragraph, p. 5. 

Reviewer #4: This manuscript presents valuable data which confirms the current knowledge regarding the natural history of renal cell carcinoma.

My comments are as follows:

The manuscript needs minor revision in terms of scientific writing.

The title talks about renal cell carcinoma but as far as I understood you have included Wilms tutor patients. I think one should be changed, either title or the analysis.

>>> 98.5% of cases are RCC, addition p. 3.

I honestly did not understand the purpose of section " Diagnostics and treatment for RCC". The fact that you mentioned this after methods part is a bit confusing.

>>> As this study covers 50 years we feel that it is important to highlight changes in diagnostic and treatment over this period which most readers do not know. We refer to this section also in Introduction, p. 3.

Reviewer #5: Studies on Renal Cell Carcinoma survival till date has shown favorable survival outcomes. However, these studies have been performed over limited periods of time. In this manuscript, authors examined survival rates in RCC over an extended period of time in Swedish and Finland populations. They analyzed RCC survival in Finland and Sweden (with similar health care systems) over a 50-year period (1967-2016) using data from the NORDCAN database provided by the local cancer registries.

In addition to the standard 1- and 5-year relative survival rates, they calculated the difference between 1 year and 5-year survival as a measure of how well survival was maintained between years 1 and 5. While relative 1-

year survival rates increased linearly in both countries and reached 90% in Sweden and 80% in

Finland, 5-year survival also developed favorably. The difference between 1- and 5-year survival rates did not improve in Sweden suggesting that the gains in 5-year survival were owing to gains in 1-year survival. In Finland there was a gain in survival between years 1 and 5, but the gain in 1-years survival was the main contributor to the favorable 5-year survival. The authors conclude that earlier diagnosis and surgical treatment of RCC is the main driver of the favorable change in survival during the past 50 years. This study reports novel findings. Though due to limited data available in NORDCAN database, there was lack of information on performed treatments and clinical stage, precluding the analyses of important potential variables.

>>> The lacking of individual data are acknowledged under limitations on p. 7.

---

## [Decision Letter · Decision Letter 1]

1 Jun 2021

PROGRESS IN SURVIVAL IN RENAL CELL CARCINOMA THROUGH 50 YEARS EVALUATED IN FINLAND AND SWEDEN

PONE-D-21-06223R1

Dear Dr. Kari Hemminki,

We’re pleased to inform you that your manuscript has been judged scientifically suitable for publication and will be formally accepted for publication once it meets all outstanding technical requirements.

Kind regards,

Wen-Wei Sung, M.D., Ph.D.

Academic Editor

PLOS ONE

Reviewers' comments:

Reviewer's Responses to Questions

**Comments to the Author**

1. If the authors have adequately addressed your comments raised in a previous round of review and you feel that this manuscript is now acceptable for publication, you may indicate that here to bypass the “Comments to the Author” section, enter your conflict of interest statement in the “Confidential to Editor” section, and submit your "Accept" recommendation.

Reviewer #1: All comments have been addressed

Reviewer #2: All comments have been addressed

Reviewer #3: All comments have been addressed

Reviewer #4: All comments have been addressed

Reviewer #5: All comments have been addressed

2. Is the manuscript technically sound, and do the data support the conclusions?

Reviewer #1: Yes

Reviewer #2: Yes

Reviewer #3: Yes

Reviewer #4: Yes

Reviewer #5: Yes

3. Has the statistical analysis been performed appropriately and rigorously? 

Reviewer #1: Yes

Reviewer #2: Yes

Reviewer #3: N/A

Reviewer #4: I Don't Know

Reviewer #5: Yes

4. Have the authors made all data underlying the findings in their manuscript fully available?

Reviewer #1: Yes

Reviewer #2: Yes

Reviewer #3: Yes

Reviewer #4: Yes

Reviewer #5: Yes

5. Is the manuscript presented in an intelligible fashion and written in standard English?

Reviewer #1: Yes

Reviewer #2: Yes

Reviewer #3: Yes

Reviewer #4: Yes

Reviewer #5: Yes

6. Review Comments to the Author

Reviewer #1: The authors have answered all the reviewers' concern about this database study. Although this study is limited because of the database design, it is still helpful for readers to see the treatment trend of mRCC.

Reviewer #2: The raised comment has been addressed adequately.

This study provide important information about survival of RCC in Finland.

Reviewer #3: (No Response)

Reviewer #4: (No Response)

Reviewer #5: Authors have satisfactorily addressed this reviewer's raised concern by modifying the discussion. The revised version is acceptable.

7. PLOS authors have the option to publish the peer review history of their article (what does this mean?). If published, this will include your full peer review and any attached files.

Reviewer #1: No

Reviewer #2: **Yes: **Shu-Pin Huang

Reviewer #3: No

Reviewer #4: No

Reviewer #5: No

---

## [Editor Report · Acceptance letter]

14 Jun 2021

PONE-D-21-06223R1 

Progress in survival in renal cell carcinoma through 50 years evaluated in Finland and Sweden 

Dear Dr. Hemminki:

I'm pleased to inform you that your manuscript has been deemed suitable for publication in PLOS ONE. Congratulations! Your manuscript is now with our production department. 

Kind regards, 

on behalf of

Dr. Wen-Wei Sung 

Academic Editor

PLOS ONE